# Gap-Aware Preference Optimization: Enhancing Model Alignment with Perception Margin

## Abstract

Reinforcement learning from human feedback (RLHF) approaches are widely used for fine-tuning large language models (LLMs) to align with instructional preferences. However, traditional RLHF methods often rely on binary labels, which fail to capture the pairwise differences in human perception, leading to potential performance degradation. To address this limitation, we introduce **Gap-Aware Preference Optimization** (GaPO), a novel approach that integrates the degree of semantic gaps into preference optimization. By modifying the existing margin term in the loss function and replacing it with an estimated gap computed using general metrics, GaPO provides a new supervisory signal that explicitly highlights the nuances between preference pairs. This new signal helps the model allocate gradients more rationally during optimization, facilitating more effective learning from the preference data. Experiments conducted with a strong base model, Llama-3-8B-Instruct, demonstrate that GaPO surpasses State-of-the-Art methods on widely used benchmarks. Our best-performing model, GaPO-ROUGE_L, achieves a win rate of 52.8% on AlpacaEval 2.0, exceeding the baseline methods by 5.3 points.

## 1 Introduction

Reinforcement Learning with Human Feedback (RLHF, Christiano et al. (2017); Ouyang et al. (2022)) has proven to be an effective and promising method for fine-tuning large-scale language models (LLMs, Achiam et al. (2023); Bai et al. (2023); Dubey et al. (2024)), aligning their generative preferences with human-established standards. This alignment extends beyond content accuracy to encompass attributes such as helpfulness, harmlessness, and logical coherence, which are fundamental to human linguistic norms (Wang et al., 2024a). Furthermore, RLHF enables the customization of generative behaviors to meet the specific demands of particular tasks. Notably, RLHF-optimized models with fewer parameters can achieve generation quality that is comparable to, or even rivals, that of larger models. This significantly enhances the practical utility of smaller models in real-world applications.

The RLHF dataset consists of winning and losing response pairs, assigned binary labels of 1 and 0, respectively. This binary labeling scheme does not reflect the nuanced quality differences between the preference pairs. During training, this results in uniform treatment of all data pairs, potentially causing a sub-optimal optimization trajectory. Specifically, the model might disproportionately focus on complex examples at the expense of adequately learning from simpler, yet informative data points, thus undermining its general fitting performance.

Alternatively, the model might expand its search space excessively to accommodate challenging cases, resulting in a potential decrease in the log probabilities of positive examples and creating unmanageable contamination of the base model.

From a human perspective, some winning cases may only be marginally better than losing cases, while others can be substantially superior. Therefore, during optimization, it is essential to allocate more gradient updates to the latter. Although the reward functions in previous Preference Optimization approaches assess the overall sentence generation probability by cumulatively summing the generation probabilities of each token, they still fail to fully consider the sentence as a whole and do not effectively compute the margin at the sentence level.

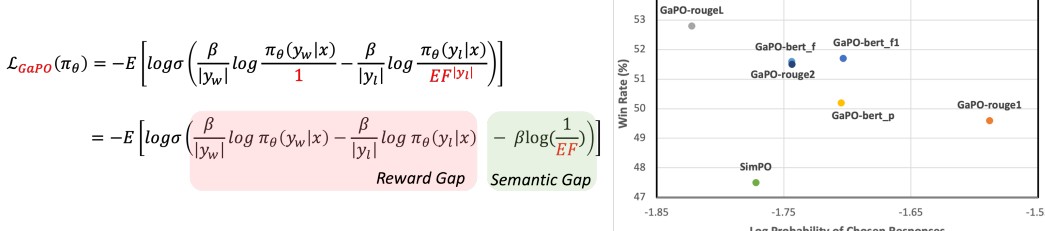

$$\mathcal{L}_{GaPO}(\pi_\theta) = -E\left[log\sigma\left(\frac{\beta}{|y_w|}log\frac{\pi_\theta(y_w|x)}{1} - \frac{\beta}{|y_l|}log\frac{\pi_\theta(y_l|x)}{EF^{|y_l|}}\right)\right]$$

$$= -E\left[log\sigma\left(\underbrace{\frac{\beta}{|y_w|}log\,\pi_\theta(y_w|x) - \frac{\beta}{|y_l|}log\,\pi_\theta(y_l|x)}_{Reward\ Gap}\ \underbrace{-\ \beta log(\frac{1}{EF})}_{Semantic\ Gap}\right)\right]$$

Figure 1: ***Left***, GaPO utilize an estimated semantic gap to instruct reward gap optimization. ***Right***, Scatter plot illustrating the average log probability of chosen responses and Win Rate on the AlpacaEval 2.0 evaluation benchmark, highlighting GaPO's capability to improve Win Rates in downstream tasks.

We first seek to quantify and simulate the perceived differences between pairs of training data with minimal complexity. To achieve this, we tested various traditional machine translation evaluation metrics, such as Jaccard Score (Costa, 2021), ROUGE (Lin (2004); Lin & Och (2004)), and BERTScore (Zhang et al., 2019).

Secondly, to incorporate these degree metrics into the training phase and reduce the model's fitting difficulty, we propose a straightforward transformation of these metrics into a "gap" score, which represents the reward difference between winning and losing examples in the loss function. Empirically, we experimented with several mappings to ensure compatibility with the reward space.

From another perspective, our novel loss function can also be viewed as a replacement for the reward function in DPO (See Figure 1). In this framework, the reward function for winning examples is assigned a denominator of 1, while the denominator for losing examples is dynamically determined by our evaluation factor ($EF$). This approach not only circumvents potential conflicts between reward optimization and the actual log probability optimization objective, but also alleviates inconsistencies in the gaps between training pairs.

In conclusion, our contributions are highlighted as follows:

- **Introduction of GaPO**: We propose GaPO, a novel method that introduces the concept of human preference intensity to provide additional preference information. This approach not only aligns with human preferences but also ensures that the model reflects the strength of these preferences. Consequently, it enhances the model's ability to accurately capture and reflect the subtleties of human preference intensity. Additionally, this approach ensures that the log probability of generating a good response does not experience a significant decrease during the training phase.

- **Comparison with State-of-the-Art Methods**: We compare our GaPO method with state-of-the-art approaches, including DPO and SimPO. Our results demonstrate that the GaPO loss function effectively utilizes pairwise gap instruction signals to achieve superior performance in downstream tasks.

- **Explore Different Gap forms**: We empirically evaluate the performance of GaPO by experimenting with different EF function forms and normalization techniques. Our model trained by ROUGE_L as estimated gap values achieves 52.8% win rate on the AlpacaEval 2.0 test set.

## 2 RELATED WORKS

Traditional RLHF (Reinforcement Learning from Human Feedback) such as PPO (Schulman et al., 2017) methods typically involve optimizing reward functions derived from human preferences. While this approach is effective, it can also introduce some challenges, such as increased computational complexity and the need to consider bias-variance trade-offs when estimating and optimizing rewards (Schulman et al., 2015).

Table 1: Summary of $EF$ Metrics and Their Calculation Formulas.

| $EF$ Metric | | Description | Calculation Formula |
|---|---|---|---|
| Jaccard Score | | The similarity between two sets by comparing the intersection and the union of the sets. | $\frac{\|y_w \cap y_l\|}{\|y_w \cup y_l\|}$ |
| ROUGE | 1 | The overlap of unigrams. | $\text{F1}\left(\frac{\|\text{Common Unigrams}\|}{\|\text{Unigrams in } y_l\|}, \frac{\|\text{Common Unigrams}\|}{\|\text{Unigrams in } y_w\|}\right)$ |
| | 2 | The overlap of bigrams. | $\text{F1}\left(\frac{\|\text{Common Bigrams}\|}{\|\text{Bigrams in } y_l\|}, \frac{\|\text{Common Bigrams}\|}{\|\text{Bigrams in } y_w\|}\right)$ |
| | L | The longest common subsequence (LCS). | $\text{F1}\left(\frac{\|\text{LCS}(y_w, y_l)\|}{\|y_l\|}, \frac{\|\text{LCS}(y_w, y_l)\|}{\|y_w\|}\right)$ |
| BERT Score | P | The precision using BERT (Devlin, 2018) embeddings to evaluate the similarity. | $\frac{\sum_i \max_j \cos(\text{BERT}y_w[i], \text{BERT}y_l[j])}{\|y_l\|}$ |
| | R | The recall using BERT embeddings to evaluate the similarity. | $\frac{\sum_j \max_i \cos(\text{BERT}y_w[i], \text{BERT}y_l[j])}{\|y_w\|}$ |
| | F1 | The F1 score using the precision and recall from BERT embeddings. | $\text{F1}(\text{BERTScore}_p, \text{BERTScore}_r)$ |

Recent research has explored other methods aimed at directly optimizing LLMs strategies based on human preferences, without relying on a scalar reward signal (Ouyang et al., 2022). The goal of these methods is to simplify the alignment process, reduce computational overhead, and achieve more robust optimization by directly utilizing preference data. The most well-known approach is DPO (Rafailov et al., 2024), which employs a method called the Bradley-Terry model. It directly optimizes preference data pairs by leveraging an analytical mapping from the reward function to the optimal policy. However, DPO is highly sensitive to the parameter beta, making it prone to overfit to preference data (Feng et al., 2024). This may reduce the probability of generating good responses, leading to sub-optimal training outcomes. Additionally, the use of a reference model causes a inconsistency between the win-loss reward ranking in the training objective and the model win-loss output ranking.

Currently, many works have focused on optimization from different perspectives, which can be mainly divided into two directions:

**Retain the Reference Model.** $\beta$-DPO (Wu et al., 2024) concentrates on data quality and trains using batch-level dynamic $\beta$ adjustments. R-DPO (Park et al., 2024) and DPOP (Pal et al., 2024) both introduce new normalization terms, represented by the difference in sentence length and the difference in generation probability from the model of optimization and reference, respectively. RSO (Liu et al., 2023) computes gradients only for data pairs where there is a discrepancy between the model-generated objectives and human preferences, and it employs a rejection sampling method to acquire preference data. WPO (Zhou et al., 2024) adjusts the weights of data pairs based on the preference output information provided by the current model policy. IPO (Azar et al., 2024) and KTO (Ethayarajh et al., 2024) use KL divergence to guide model updates and IPO ensures that the KL regularisation remains effective.

**Remove the Reference Model.** RRHF (Yuan et al., 2023) trains by using rank loss robustly without complex hyperparameter tuning. SLiC-HF (Zhao et al., 2023) calculates contrastive loss through Sequence Likelihood Calibration and introduces a regularization term to increase the margin distance. ORPO (Hong et al., 2024) propose a reference-free loss function that enables simultaneous supervised fine-tuning and preference alignment within a single training session. CPO (Xu et al., 2024) directly uses log likelihood as a reward function and the SFT objective as a regulation. SimPO (Meng et al., 2024) introduces a target reward margin $\gamma$, which helps to separate winning and losing responses. Additionally, most methods incorporate SFT loss to ensure the probability of generating high-quality responses, and they all use sequence length as a normalization factor. On a higher level, GPO (Tang et al., 2024) summarizes preference optimization from a unified perspective.

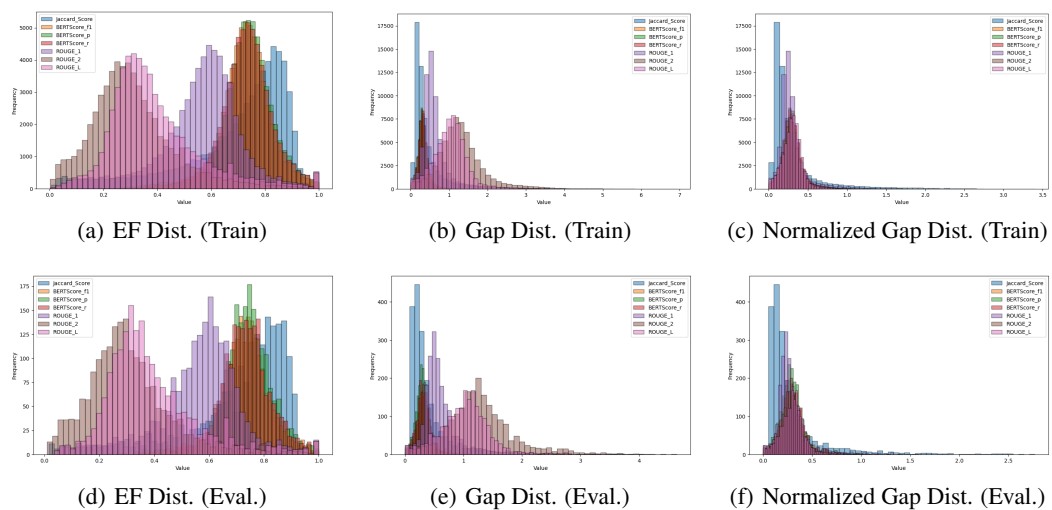

(a) EF Dist. (Train)  (b) Gap Dist. (Train)  (c) Normalized Gap Dist. (Train)

(d) EF Dist. (Eval.)  (e) Gap Dist. (Eval.)  (f) Normalized Gap Dist. (Eval.)

Figure 2: Comparative Analysis of Estimated Distributions for Different $EF$ Forms in the Llama3-Ultrafeedback-Armorm Dataset. This figure illustrates the distributions of various $EF$ forms used to compute preference pairs. Through the logarithmic mapping and normalization, we attain a more compact distribution, which is optimally configured to define the reward space for effective fine-tuning guidance.

## 3  GaPO: GAP AWARE PREFERENCE OPTIMIZATION

### 3.1  RETHINKING ABOUT DPO AND SIMPO

Direct preference optimization (DPO, Rafailov et al. (2024)) is currently one of the most widely used methods for model preference alignment. Given a triplet $(x, y_w, y_l)$ from a preference training dataset $\mathcal{D}$ consisting of the prompt $x$, the winning response $y_w$, and the losing response $y_l$, the objective of DPO is to maximize the log-likelihood of $p(y_w > y_l|x)$ by direct policy optimization without explicit reward estimation.

Based on the Bradley-Terry model and the reparameterized reward functions, the loss function can be derived as the following form

$$\mathcal{L}_{\text{DPO}}(\pi_\theta; \pi_{\text{ref}}) = -\mathbb{E}_{(x, y_w, y_l) \sim \mathcal{D}} \left[ \log \sigma \left( r_{DPO}(x, y_l) - r_{DPO}(x, y_l) \right) \right], \quad (1)$$

where $r_{DPO}(x, y) = \beta \log \frac{\pi_\theta(y|x)}{\pi_{\text{ref}}(y|x)}$. By performing some algebraic manipulations, we obtain a margin-based form of DPO,

$$\mathcal{L}_{\text{DPO}}(\pi_\theta; \pi_{\text{ref}}) = -\mathbb{E}_{(x, y_w, y_l) \sim \mathcal{D}} \left[ \log \sigma \beta \left( \log \pi_\theta(y_w \mid x) - \log \pi_\theta(y_l \mid x) - \gamma \right) \right], \quad (2)$$

where $\gamma = log\pi_{\text{ref}}(y_w \mid x) - log\pi_{\text{ref}}(y_l \mid x)$ is a margin term. Recent advancements in SimPO (Meng et al., 2024) demonstrate that in DPO, satisfying the reward ranking $r_{DPO}(x, y_w) > r_{DPO}(x, y_l)$ does not necessarily imply that the likelihood ranking $p_\theta(y_w \mid x) > p_\theta(y_l \mid x)$ is met. From the perspective of margins, this means $\gamma$ is not always positive.

To mitigate the impact of optimization inconsistency, SimPO replaces the reward formulation in DPO with the length normalized $p_\theta$ to align with the nature of maximized log-likelihood of sequences in LLM inference and apply a fixed positive hyper-parameter $\gamma$ as margin, yielding the following form:

$$\mathcal{L}_{\text{SimPO}}(\pi_\theta) = -\mathbb{E}_{(x, y_w, y_l) \sim \mathcal{D}} \left[ \log \sigma \left( \frac{\beta}{|y_w|} \log \pi_\theta(y_w|x) - \frac{\beta}{|y_l|} \log \pi_\theta(y_l|x) - \gamma \right) \right], \quad (3)$$

While SimPO effectively establishes a discrepancy between reward and generation for preference optimization, it still potentially sacrifices pairwise optimization by employing a fixed margin $\gamma$ compared to DPO. Some training pairs exhibit a clear distinction from a human perception perspective, whereas others are merely borderline cases. Addressing these pairs with the same preset margin could lead to an unnecessary compromise of distinctly identifiable cases when fitting borderline ones.

## 3.2 GaPO Objective

Accurate preference training in models necessitates a nuanced understanding of human perception. To address this issue, it is crucial to develop methods for computing and stimulating the degrees of human perception. By quantifying perception, we can introduce a spectrum of preference intensities that provide additional layers of information beyond binary labels. This enriched data allows models to differentiate between varying degrees of preference quality, leading to a more refined and accurate optimization process.

**Utilizing estimated margin into Preference optimization.** Intuitively, we directly employ a pairwise margin term to introduce a gap-related signal,

$$\mathcal{L}_{\text{GaPO}}\left(\pi_\theta\right) = -\mathbb{E}_{(x,y_w,y_l)\sim\mathcal{D}}\left[\log\sigma\left(\beta\left(\Delta\hat{r} - \text{Estimated Margin}\right)\right)\right] \tag{4}$$

Where $\Delta\hat{r} = \hat{r}_{y_w} - \hat{r}_{y_l}$ represents the difference in the values of the implicit reward functions.

We adopted SimPO's reward formulation as $\hat{r}$ because it directly optimizes log-likelihood and normalizes the length of response, resulting in the following specific form:

$$\mathcal{L}_{\text{GaPO}}\left(\pi_\theta\right) = -\mathbb{E}_{(x,y_w,y_l)\sim\mathcal{D}}\left[\log\sigma\left(\frac{\beta}{|y_w|}\log\pi_\theta\left(y_w\mid x\right) - \frac{\beta}{|y_l|}\log\pi_\theta\left(y_l\mid x\right)\right.\right.$$
$$\left.\left. - \beta\text{Estimated Margin}\right)\right] \tag{5}$$

**Estimate the Human Perception Gap.** To quantify the superiority between pairs of data in human perception, we selected evaluation metrics commonly used in the field of machine translation, including Jaccard Score, ROUGE, and BERTScore (See table 1 for more details). These metrics are not only simple to compute and cost-effective but also effectively capture the degree-based characteristics of the data. We collectively term these metrics as the Evaluation Factor ($EF$).

$EF$ measures the gap between the winning response and the losing response. Specifically, an $EF$ value closer to 1 denotes a pair of responses with minimal difference, signifying a close match, whereas a value nearing 0 signifies a larger gap. To utilize it in the margin term, we choose a negative logarithmic mapping, Estimated Margin $= -\log(EF)$, then we have

$$\mathcal{L}_{\text{GaPO}}\left(\pi_\theta\right) = -\mathbb{E}_{(x,y_w,y_l)\sim\mathcal{D}}\left[\log\sigma\left(\frac{\beta}{|y_w|}\log\pi_\theta\left(y_w\mid x\right) - \frac{\beta}{|y_l|}\log\pi_\theta\left(y_l\mid x\right) - \beta\log(\frac{1}{EF})\right)\right] \tag{6}$$

$$\Rightarrow -\mathbb{E}_{(x,y_w,y_l)\sim\mathcal{D}}\left[\log\sigma\left(\frac{\beta}{|y_w|}\log\frac{\pi_\theta\left(y_w\mid x\right)}{1} - \frac{\beta}{|y_l|}\log\frac{\pi_\theta\left(y_l\mid x\right)}{EF^{|y_l|}}\right)\right]. \tag{7}$$

Since the $EF$ always ranges between $(0,1)$, the estimated margin term is always positive, which helps the model effectively distinguish between winning and losing responses in the correct direction. From the perspective of an implicit reward function, the reward ranking still guarantees the likelihood ranking. This is because $EF^{|y_l|}$ is less than 1, which assigns a larger reward to the losing response compared to the winning response.

**Normalization of the Estimated Gap.** The $EF$ distributions are closely related to the margin term in the loss function, but they vary across different metrics (see Figure 2) and cannot directly fit

Table 2: Experiment results compared with state-of-the-art fine-tuning methods. We use Llama3-Instruct(8B) as the base model and an enhanced preference dataset ranked by a strong reward model ArmoRM-Llama-8B-v0.1. We report performance on commonly used benchmarks, including AlpacaEval 2.0, Arena-Hard, and MT-Bench.

| Method | | Llama3-Instruct(8B) | | | | | |
|---|---|---|---|---|---|---|---|
| | | AlpacaEval 2.0 | | | Arena-Hard | | MT-Bench |
| | | LC(%) | WR(%) | Length | WR(%) | Length | GPT-4 Turbo |
| SFT | | 26.0 | 25.3 | 1920 | 22.3 | 596 | 6.9 |
| RLHF | RRHF | 37.9 | 31.6 | 1700 | 28.8 | 467 | 7.1 |
| | SLiC-HF | 33.9 | 32.5 | 1938 | 29.3 | 599 | 6.9 |
| | DPO | 48.2 | 47.5 | 2000 | 35.2 | 609 | 7.0 |
| | IPO | 46.8 | 42.4 | 1830 | **36.6** | 527 | 7.2 |
| | CPO | 34.1 | 36.4 | 2086 | 30.9 | 604 | **7.2** |
| | KTO | 34.1 | 32.1 | 1878 | 27.3 | 541 | **7.2** |
| | ORPO | 38.1 | 33.8 | 1803 | 28.2 | 520 | **7.2** |
| | R-DPO | 48.0 | 45.8 | 1933 | 35.1 | 608 | 7.0 |
| | SimPO | 53.7 | 47.5 | 1777 | 36.5 | 530 | 7.0 |
| | GaPO-ROUGE_L | **56.1** | **52.8** | 1,902 | 36.3 | 538 | 7.1 |

the reward space. To obtain a more stable estimated gap, we apply normalization using a hyperparameter $\gamma$. Specifically, we have adopted a scaling normalization approach, which means

$$\text{Normalized Margin} = Norm(\log \frac{1}{EF}, \gamma) = \gamma \log(\frac{1}{EF})/\overline{\log(\frac{1}{EF})} \tag{8}$$

After the logarithmic mapping and normalization, we observe a more compact distribution that preserves the ranking and degree of the original $EF$. Finally, we get the Loss funtion of GaPO,

$$\mathcal{L}_{\text{GaPO}}(\pi_\theta) = -\mathbb{E}_{(x,y_w,y_l)\sim\mathcal{D}}\left[\log \sigma \left(\frac{\beta}{|y_w|}\log \pi_\theta(y_w \mid x) - \frac{\beta}{|y_l|}\log \pi_\theta(y_l \mid x)\right.\right.$$
$$\left.\left. - Norm(\log \frac{1}{EF}, \gamma)\right)\right] \tag{9}$$

## 4 EXPERIMENTS

In this section, we conduct extensive experiments to show the performance of our GaPO method and compare it with other baselines. We further undertake a sequence of ablation studies to illustrate the impact of the metrics for $EF$, different mapping functions, and normalization forms. Additionally, we provide a qualitative assessment focusing on log probability metrics and models' performance on the downstream task.

### 4.1 EXPERIMENTAL SETUP

**Base Model and Dataset.** We follow the experimental configuration as demonstrated by SimPO. Specifically, we leverage the instruct-tuned model, meta-llama/Meta-Llama-3-8B-Instruct (Dubey et al., 2024), as our foundational model, alongside the princeton-nlp/llama3-ultrafeedback-armorm (Cui et al. (2023), Meng et al. (2024)) dataset for training. This dataset is crafted using RLHFlow/ArmoRM-Llama3-8B-v0.1 (Wang et al., 2024b) as the reward model to evaluate and prioritize the generated data, thereby establishing a superior and highly resilient preference dataset.

Table 3: Exploring the Impact of Various $EF$ Metrics on Performance. This ablation study investigates the effectiveness of different machine translation evaluation metrics—such as Jaccard Score, BERTScore, and ROUGE when computing the EF. Our findings indicate that ROUGE scores exhibit the highest performance.

| Method | | | Llama3-Instruct(8B) | | | |
| --- | --- | --- | --- | --- | --- | --- |
| | | | AlpacaEval 2.0 | | | MT-Bench |
| | | | LC(%) | WR(%) | Length | GPT-4 Turbo |
| **SFT** | | | 26.0 | 25.3 | 1920 | 6.9 |
| RLHF | DPO | | 48.2 | 47.5 | 2000 | 7.0 |
| | SimPO | | 53.7 | 47.5 | 1777 | 7.0 |
| | GaPO | Jaccard_Score | 51.0 | 44.4 | 1745 | 7.0 |
| | | BERTScore_f1 | 55.2 | 51.7 | 1,888 | 7.0 |
| | | BERTScore_r | 55.0 | 51.6 | 1,906 | 7.1 |
| | | BERTScore_p | 54.6 | 50.2 | 1,856 | 7.0 |
| | | ROUGE_1 | 53.4 | 49.6 | 1,859 | 7.1 |
| | | ROUGE_2 | 55.1 | 51.5 | 1,884 | **7.3** |
| | | ROUGE_L | **56.1** | **52.8** | 1,902 | 7.1 |

The hyper-parameters are consistent with those used in SimPO, we set learning rate $= 1e^{-6}$, $\beta = 10$ and $\gamma = 0.3$.

**Benchmarks.** Following previous works, we use AlpacaEval 2.0 (Dubois et al., 2024), MT-Bench (Zheng et al., 2023), and Arena-Hard (Li et al., 2024) as our evaluation benchmarks.

- **AlpacaEval 2.0** is an LLM-based automatic evaluation benchmark. It utilizes the Alpaca-Farm dataset, which comprises a diverse set of general human instructions as prompts. The benchmark evaluates model responses by comparing them with reference responses generated by GPT-4-Turbo. These comparisons are conducted using a GPT-4-Turbo-based annotator. Following standard evaluation procedures, we report both the raw win rate (WR) and the length-controlled win rate (LC) of model responses over the reference responses.

- **MT-Bench** is a collection of 80 high-quality multi-turn open-ended questions. The questions cover topics like writing, role-playing, math, coding, etc.. The generated answer is judged and given a score directly without pairwise comparison, range from 0 to 10. We report the average score with GPT-4-Turbo as the judgement model.

- **Arena-Hard** is an advanced version of the MT-Bench, incorporating 500 meticulously designed technical problem-solving queries derived from challenging clusters. This benchmark employs GPT-4-Turbo as an evaluator to compare the responses of various models against a baseline model, categorizing outcomes into big win, small win, tie, small loss, and big loss. We report the win rate with GPT-4-0314 serving as the baseline model.

**Baselines.** We compare our method with various offline preference optimization methods, including RRHF (Yuan et al., 2023), SLiC-HF Zhao et al., 2023, DPORafailov et al. (2024), IPO(Azar et al., 2024), CPO (Xu et al., 2024), KTO(**?**), ORPO (Hong et al., 2024), R-DPO(Park et al., 2024), and SimPO(Meng et al., 2024). The proposed methods aim to align closely with human preferences through varied objectives. However, they generally overlook the potential of utilizing the human perception gap for fine-tuning enhancements. SimPO is the most closely related baseline, and we utilize its length normalization form implicit reward function. Notably, our GaPO method is compatible with most DPO-based approaches (beyond just SimPO), further enhancing its applicability.

Table 4: Ablation Study on the Efficacy of Mapping Functions and Normalization Approaches. We investigated a range of mapping functions designed to inversely convert $EF$ into an estimated gap value, finding that the logarithmic mapping function outperforms others. Furthermore, our experimentation with the absence of normalization and an alternative additional normalization highlights the advantageous performance of scaling normalization.

| Method | | | AlpacaEval 2.0 | | |
|---|---|---|---|---|---|
| | | | LC(%) | WR(%) | Length |
| DPO | | | 48.2 | 47.4 | 2000 |
| SimPO | | | 53.7 | 47.5 | 1777 |
| GaPO Rouge_L | $1 - EF$ | | 53.9 | 50.9 | 1910 |
| | $1 - EF^2$ | | 53.9 | 50.7 | 1901 |
| | $1 - \sqrt{EF}$ | | 55.0 | 52.0 | 1915 |
| | $\log(\frac{1}{EF})$ | w/o norm | 28.8 | 25.4 | 1717 |
| | | add. norm | 53.5 | 50.2 | 1898 |
| | | scale. norm (ours) | **56.1** | **52.8** | 1902 |

## 4.2 EXPERIMENT RESULTS AND ABLATION

**GaPO significantly improves performance on AlpacaEval 2.0.** As shown in Table 3.2, our GaPO method achieves the best performance on the AlpacaEval 2.0 evaluation dataset. Specifically, GaPO attains a notable win rate (WR) of 52.8% and a length-controlled win rate (LC) of 56.1%, outperforming the best baseline, SimPO, by 5.3 and 2.4 percentage points, respectively. In the more challenging benchmark Arena-Hard, our GaPO method performs comparably to the baseline SimPO in terms of win rate. This suggests that GaPO does not demonstrate an enhancement over SimPO in addressing complex problems, potentially owing to constraints in the quality of the dataset and the capacity of the model. In the MT-Bench benchmark, our GaPO method attains a score of 7.1, marginally outperforming the SimPO score of 7.0. However, the MT-Bench benchmark demonstrates limited discriminatory capacity when assessing diverse responses across the three datasets. This limitation could stem from the assessment model's dependence on single-score evaluations, which tend to be less sensitive at discerning fine-grained distinctions than pairwise comparison methods.

**Different EF metrics show different improvements.** We explored various functions to compute the evaluation factor (EF) for assessing human perception of the gap between pairs, including Jaccard Score, BERTScore_R, BERTScore_P, BERTScore_F1, ROUGE-1, ROUGE-2, and ROUGE-L. Results can be found in table 1. BERTScore and ROUGE metrics improved the win rate from 4.2 points to 5.3 points compared to the baselines DPO and SimPO on AlpacaEval 2.0. For the MT-Bench dataset, ROUGE-2 achieved a score of 7.3 compared to 7.0 for SimPO and DPO. However, the Jaccard Score and ROUGE-1 show relatively poor performance, indicating that they may not accurately reflect the true gap between the reference and the candidate responses due to a lack of semantic information.

**Logarithmic mapping and caling normalization are the most effective.** In table 4, we report the experiment results with different functional forms to map the evaluation factor ($EF$) within the range of 0 to 1 to an estimated gap value. The forms we evaluated included Estimated Gap $= 1 - EF$, Estimated Gap $= 1 - EF^2$, and Estimated Gap $= 1 - \sqrt{EF}$. Compared to the log form Estimated Gap $= \frac{1}{EF}$ using ROUGE-L as the $EF$, we observed a decrease in win rate ranging from 0.8 points to 1.8 points. Removing the normalization had the most negative impact on the results, leading to an almost 50% decrease in performance. We also tested an additional form of normalization:

$$\text{Norm}\left(\log \frac{1}{EF}, \gamma\right) = \log\left(\frac{1}{EF}\right) - \overline{\log\left(\frac{1}{EF}\right)} + \gamma, \tag{10}$$

which resulted in a decrease in win rate by 2.6 points, indicating that the scaling normalization is the most effective.

## 4.3 QUALITATIVE ANALYSIS

In preference optimization, our goal is to distinguish between preference pair rewards while maintaining a high log probability for the winning response. This approach helps prevent significant alterations compared to the base model during fine-tuning, which could result in unexpected outcomes during inference. Often, a decline in log probability occurs because, during the fine-tuning process, the model explores a larger search space to optimize the reward difference between the winning response and the losing response, aiming to meet the preset gap in the loss function. The goal of GaPO is to optimize this preset gap for each pair of training data, fitting the data better and achieving higher win rates in downstream tasks without substantial adjustments to the base model parameters. In the figure 1, we observe that the GaPO method achieves better performance in downstream tasks compared to SimPO, while maintaining a log probability similar to or even higher than that of SimPO, highlighting the superiority of the GaPO method.

## 5 DISCUSSION AND FUTURE WORK

**Conclusion.** In this work, we propose a method called GaPO, which introduces semantic gap information into the loss function. This enables the model not only to differentiate between good and bad responses but also to develop a more nuanced understanding of the degree of quality at the semantic level. This improvement aids in optimizing the gradient update process, thereby enhancing the effectiveness of RLHF. By incorporating sentence-level gap information, the model is able to reduce the log probability for the chosen response to a lesser degree while achieving a higher win rate. Additionally, our GaPO approach is designed to be interoperable with all margin-based preference optimization techniques to further improve performance.

**Future work.** Firstly, the test datasets used in this study are all derived from the question-answering (QA) domain. Given the widespread application of AI agents and the increasing demand for such technologies, future work will explore how the GAPO method can be applied within AI agent contexts, particularly in scenarios where gaps in API calls are easier to quantify. Secondly, the current semantic gap primarily relies on machine translation evaluation metrics such as ROUGE and BERTScore. Future research will explore more appropriate evaluation functions and consider using the reward model employed during DPO dataset generation to replace the existing semantic gap calculation method. Lastly, drawing from the inference generation strategy of OpenAI-o1, we plan to use Monte Carlo Tree Search (MCTS) in future work to generate datasets, which will be combined with GAPO training to further optimize model performance and its ability to capture human preferences.

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
