# A  APPENDIX

## A.1  GRADIENT ANALYSIS

We conduct a analysis of the gradients in GAPO with the aim of examining the specific influence of the $EF$ factor during the gradient update process.

$$\nabla\mathcal{L}_{\text{GaPO}} = -\beta\mathbb{E}_{(x,y_w,y_l)\sim\mathcal{D}}\Bigg[\underbrace{\sigma(\hat{r}_\theta(x,y_l) - \hat{r}_\theta(x,y_w) + \beta * log(\frac{1}{\textbf{EF}}))}_{\text{higher weight when reward estimate is wrong}}$$

$$* \Bigg[\underbrace{\nabla_\theta \log \pi(y_w \mid x)}_{\text{increase likelihood of } y_w} - \underbrace{\nabla_\theta \log \pi(y_l \mid x)}_{\text{decrease likelihood of } y_l}\Bigg]\Bigg] \quad (1)$$

For optimal analysis, the $EF$ is directly substituted by the ROUGE scores. ROUGE measures the similarity between $y_w$ and $y_l$ from the perspective of recall. Therefore, when a pair of data exhibits higher content similarity, the ROUGE score increases, and the corresponding gradient decreases. This suggests that the greater the similarity between two sentences from a human perspective, the less gradient is required for model updates. Conversely, if the similarity between two sentences is perceived as lower from a human perspective, more gradient is necessary for updates. The EF provides a measure of the variance in human preference intensity, which facilitates the alignment of the model's preferences with those of humans. This alignment enhances the model's ability to reflect the intensity of preference.