# OpenReview forum: "Gap-Aware Preference Optimization: Enhancing Model Alignment with Perception Margin"
_ICLR.cc/2025/Conference — ICLR 2025 Conference Withdrawn Submission_

### Official Review · Reviewer_J2tT · 2024-10-31

**Soundness:** 2
**Presentation:** 3
**Contribution:** 3
**Rating:** 5
**Confidence:** 5

**Summary:**

This paper proposes a new approach for preference optimization named Gap-Aware Preference Optimization (GaPO), which leverages sematic gaps as a signal for adjusting the gradient during training. The GaPO builds upon SimPO training objective by introducing an extra evaluation factor (EF) as regularization. Ablation studies are conducted on AlpacaEval 2.0, Arena-Hard and MT-Bench to prove the effectivessness of the proposed GaPO training objective.

**Strengths:**

The paper proposes integrating the measurement of the semantic gap between chosen and rejected responses into the training objective for preference optimization. Specifically, a semantic gap is added to SimPO as a regularization term to create a training objective that accounts for the semantic gaps in preference pairs.

**Weaknesses:**

- The experimental results in Tab. 2 do not show a good performance of GaPO. Consider that GaPO is an improved version of SimPO, GaPO do not outperform SimPO on Arena-Hard, and the improvement on MT-Bench is marginal. Although the win rate of GaPO on AlpacaEval 2.0 is better, the response of GaPO is lengthy. Consider that LLM judges often favor longer responses (even with LC), I do not believe the results can prove its superiority against SimPO based solely on AlpacaEval 2.0. Similarly, in Fig. 1 (right) and Tab. 2, although GaPO achieves a higher LC win rate in some cases, the overall response length is longer than that of SimPO.

- The author(s) claim that the uniform treatment of preference pairs (regardless of the differences between responses) expands the search space during training, potentially resulting in a decrease in log probabilities. However, as shown in Fig. 1 (right), GaPO-rougeL achieves the best performance among different methods, while also causing a decrease in log probability, which contrasts with the idea of limiting the search space by accounting for the senmatic gaps between preference pairs. Insead, the results show that across different EF metrics, the win rate decreases with higher log probability.

**Questions:**

- The author(s) might provide additional evidence or explanations to demonstrate the superiority of GaPO.

- The results on Arena-Hard benchmark are not provided in Tab. 3 and 4. Given that Arena-Hard offers better separability and alignment with human preferences, the author(s) might consider including these results.

- The current explanations for the search space and log probability require reconsideration. The authors might need to address the inconsistency between the theoretical explanation for search space and log probabilities and the experimental results.

---

### Official Review · Reviewer_9L4e · 2024-11-01

**Soundness:** 2
**Presentation:** 3
**Contribution:** 2
**Rating:** 5
**Confidence:** 3

**Summary:**

This paper identifies the binary format of RLHF data labels fails to reflect the actual pairwise difference of human preference. In order to provide supervision signals beyond binary labels, this work proposes to weight the pairwise samples with respect to the semantic gap. This approach achieves competitive performance on multiple benchmarks.

**Strengths:**

- The paper presents its motivation, methodology, results clearly.
- The proposed method is simple, and has a clear connection to related works
- The work obtains significant advantage on Alpaca Eval 2.0

**Weaknesses:**

My major concern is that the motivation is to identify perception margin, which doesn't quite match the actual implementation of Estimated Margin. The three machine translation (L247) metrics, Jaccard score, ROUGE, and BERT score, are all defiend to operate at token/multi-token level, without actually capturing the overal semantic meaning.

The empirical study shows that GaPO outperforms baselines on AlpacaEval 2.0 but not for Arena-Hard or MT-Bench. Since AlpacaEval 2.0 has a specific emphasis on the length of responses, I would speculate that all the token-level EF metrics are effective only on a narrower range of scenarios. Specifically, only when the plausible responses are largely similar at token-level can the proposed metrics be seen as proxy semantic measures. For challenging queries that expect complex responses with certain reasoning capability, token-level comparison doesn't fully capture human pairwise preference anymore.

The data-dependent margin and beyond-binary motivation is a good direction for preference optimization, however what the paper present doesn't fully exploit the potential. For example, there can be a whole much wider spectrum of margins that more faithfully match the motivation:
- Simpler variants include the length difference between $y_w$ and $y_l$;
- Semantic-level variants may consider something like response-level LM embedding distance between $y_w$ and $y_l$.

A more comprehensive study over these options may bring in further contribution in this direction, and also potentially address the non-improvement for Arena-Hard and MT-Bench.

**Questions:**

- In Section 4.3, it says "GaPO method achieves better performance in downstream tasks compared to SimPO, while maintaining a log probability similar to or even higher than that of SimPO, highlighting the superiority of the GaPO method". On the other hand, in Figure 1, most GaPO results are having a higher-than-SimPO log probability, while the "recommended" setting, GaPO-rougeL has a lower-than-SimPO log probability. How do we understand this discrepancy?
- L373 has citation format issue for KTO.

---

### Official Review · Reviewer_oM4J · 2024-11-04

**Soundness:** 2
**Presentation:** 2
**Contribution:** 2
**Rating:** 3
**Confidence:** 4

**Summary:**

In this work, the authors propose Gap-Aware Preference Optimization (GaPO), a novel approach that integrates the degree of semantic gaps into preference optimization. Results on different datasets show that GaPO could surpass existing methods, including DPO and SimPO.

**Strengths:**

1. Results on different datasets show that GaPO could surpass existing methods, including DPO and SimPO.
2. The writing is clear and easy to understand.

**Weaknesses:**

1. The effectiveness of EF is unclear. The results in Table 4, particularly the performance of GaPO without norm, do not demonstrate substantial improvement. Specifically, the LC of SFT is 26.0% and GaPO without norm is 28.8%. When GaPO is combined with the scale norm, a hyperparameter γ is introduced, which is considered the contribution from SimPO, making it difficult to evaluate the independent validity of EF. The authors are advised to conduct more experiments to demonstrate the effectiveness of EF, such as comparing GaPO with and without EF.
2. Missing gradient analysis for the EF term. The authors do not provide an analysis of the role of the EF term when applying gradient updates, for example, how the likelihood of preferred samples increases and that of dispreferred samples decreases. Including such an analysis would strengthen the paper, similar to the approach taken in Section 4 in DPO.
3. The concept of the "semantic gap" is not sufficiently clarified in the abstract. The readers may struggle to understand its meaning. It would be beneficial for the authors to provide a clear definition or an illustrative example to aid comprehension.
4. Missing Reference for KTO. In Section 4.1, the authors do not provide an appropriate reference for KTO.

**Questions:**

Please see the weaknesses above.

---

### Official Review · Reviewer_oBv2 · 2024-11-07

**Soundness:** 3
**Presentation:** 3
**Contribution:** 3
**Rating:** 6
**Confidence:** 2

**Summary:**

The paper proposed a novel approach in RLHF introducing a gap-aware margin into the preference optimization process. This approach moves beyond binary preference labels by leveraging estimated semantic gaps between responses. The proposed GaPO model effectively improves alignment with human perception by incorporating a margin term in the loss function that reflects perceived differences in quality between response pairs. Experiments on Llama-3-8B proven the effectiveness of the GaPO methods across a number of RLHF benchmarks.

**Strengths:**

- The proposed method GaPO is novel approach, it provides a alternative to binary preference optimization, by enhancing the model’s ability to accurately capture and reflect the subtleties of human preference intensity.
- The GaPO model outperformed existing approaches like SimPO and DPO on multiple benchmarks, particularly AlpacaEval 2.0, showing real-world efficacy
- The paper also explore different gap forms in GaPO by experimenting with different EF function forms and normalization techniques.

**Weaknesses:**

- It's unclear how Different EF metrics such as the one based on Jaccard Score and ROUGE-1 would perform poorly.
- Some typos: Line 373: KTO(?),

**Questions:**

(a) Could GaPO also be integrated with other RLHF methods with listwise preference such as LiPO [1]?

[1] LiPO: Listwise Preference Optimization through Learning-to-Rank, https://arxiv.org/abs/2402.01878

---

### Note · Authors · 2024-11-19

I have read and agree with the venue's withdrawal policy on behalf of myself and my co-authors.